# Structural and Electromagnetic Signatures of Anatase and Rutile NTs and Sheets in Three Different Water Models under Different Temperature Conditions

**DOI:** 10.3390/ijms241914878

**Published:** 2023-10-04

**Authors:** Eduardo Patricio Estévez Ruiz, Saravana Prakash Thirumuruganandham, Joaquín Cayetano López Lago

**Affiliations:** 1Centro de Investigación de Ciencias Humanas y de la Educación (CICHE), Universidad Indoamérica, Ambato 180103, Ecuador; eduardo45pr@gmail.com; 2Grupo de Polímeros, Departamento de Física y Ciencias de la Tierra, Escuela Universitaria Politécnica, Universidade da Coruña, 15471 Ferrol, Spain; joaquin.lopez@udc.es

**Keywords:** antase, rutile, radial distribution function (RDF), Matsui and Akaogi, PDOS, low frequency, power spectra, infrared spectra, soft phonon effect, nanotubes, nanosheets, LAMMPS MD

## Abstract

Experimental studies of TiO_2_ nanotubes have been conducted for nearly three decades and have revealed the remarkable advantages of this material. Research based on computer simulations is much rarer, with research using density functional theory (DFT) being the most significant in this field. It should be noted, however, that this approach has significant limitations when studying the macroscopic properties of nanostructures such as nanosheets and nanotubes. An alternative with great potential has emerged: classical molecular dynamics simulations (MD). MD Simulations offer the possibility to study macroscopic properties such as the density of phonon states (PDOS), power spectra, infrared spectrum, water absorption and others. From this point of view, the present study focuses on the distinction between the phases of anatase and rutile TiO_2_. The LAMMPS package is used to study both the structural properties by applying the radial distribution function (RDF) and the electromagnetic properties of these phases. Our efforts are focused on exploring the effect of temperature on the vibrational properties of TiO_2_ anatase nanotubes and an in-depth analysis of how the phononic softening phenomenon affects TiO_2_ nanostructures to improve the fundamental understanding in different dimensions and morphological configurations. A careful evaluation of the stability of TiO_2_ nanolamines and nanotubes at different temperatures is performed, as well as the adsorption of water on the nanosurface of TiO_2_, using three different water models.

## 1. Introduction

During the last few years, the extraordinary physicochemical properties of titanium dioxide (TiO${}_{2}$) nanoparticles have aroused the interest of leading researchers in the scientific community and have enabled a breakthrough in major technological applications [1]. TiO${}_{2}$ nanoparticles have been the subject of extensive research due to their remarkable optical properties, which have led to their application in various fields of technology, including photocatalysis [2,3,4], sensors [5,6], Li${}^{+}$ and H${}_{2}$ storage [7,8], rechargeable batteries [9], biomedical applications [10], photovoltaic power generation [2] and in numerous industrial products, such as paints, sunscreens, cosmetics, food (as a colourant) [11], toothpaste and even pharmaceuticals [12]. In the face of this diversity of application, major concerns have arisen about the serious thermal challenges in the fabrication and utilization of TiO${}_{2}$ nanotubes (TNTs), resulting from localized electrical, optical, and mechanical overheating. While there have been some studies of material thermal behavior, structure degradation, efficiency decrease, and mechanical stress failures, there have been few studies of the heat propagation in TNTs that would allow a proper description of the thermal performance of TNTs and an understanding of the thermophysical properties of such TNTs. Indeed, the study of the PDOS in TNTs is important to understand the propagation of heat in these materials because heat transfer through a material depends to a large extent on the material’s ability to store and transfer heat, so the thermal energy through atomic vibrations and the infrared spectrum allow to measure the absorption of energy as a function of the frequency of the infrared radiation. Faced with this research challenge, the experimental path is complex and costly, while the computational approach appears as a viable and much cheaper alternative [13] for the study of these properties in TNTs.

However, it was not until the last century that the theory of lattice dynamics based on quantum theory was established [14]. It revealed that the vibration of atoms in a crystal is correlated, and a collective vibration forms a wave of the allowed wavelength and amplitude. The quantum of such a lattice vibration is called a phonon, while phonon scattering gives the wave vector the dependence of the phonon frequencies. Experimentally, phonon scattering can be determined indirectly through the interaction between lattice waves and other waves, such as inelastic neutron scattering or electron energy loss spectroscopy [15]. Alternatively, phonon scattering can also be determined by theoretical calculation. In this regard, there are mainly two approaches based on first-principle calculations within the density functional theory [16]: the direct (frozen phonon) method and the density functional perturbation theory (DFPT). The direct method imposes the displacement of atoms and constructs the dynamical matrix based on the induced Hellman–Feynman forces [16,17,18]. This method is relatively easy to implement and, therefore, is widely used. However, such calculations generally require a supercell that should be comfortably larger than the interatomic force of the largest range, and the phonon wave vector should be proportional to the structure. In contrast, with numerous atoms and electrons in a large supercell, the calculations can become extremely long, especially as the calculating time frequently scales as the cube of the number of electrons [19].

The DFPT approach makes use of linear response theory to evaluate the dynamic matrix [20,21,22], which is considered to be quite accurate and efficient. Despite this, methods based on first-principle calculations are generally computationally demanding and often prohibitively expensive, leaving methods based on empirical or semi-empirical force fields as more or less practical options. For force field-based methods, one generally constructs the dynamical matrix by evaluating the second derivatives of the force field with respect to equilibrium positions [16,23,24], and then calculates the phonon frequencies by obtaining the eigenvalues of the dynamical matrix: traditional lattice dynamics (LD). In general, the LD provides a simple scheme to investigate phonon scattering, whenever the force field is available. However, there are also conditions where the force field formalism is too complicated to derive the Hessian, such as the charge transfer model, where a self-consistent procedure is needed [25,26]. More importantly, all the above calculation methods suffer from the fact that the dynamic matrix can only be measured at zero temperature. That is, the effect of temperature on phonons, or anharmonicity, cannot be considered directly. Instead, they usually employ a quasi-harmonic procedure to account for the finite temperature effect, which is, however, tedious and sometimes infeasible [27]. An alternative that could overcome such a deficiency is to evaluate phonons within MD simulation, where the equations of motion are integrated to track in time the positions and velocities of all atoms. MD is not limited to harmonic forces, and large amplitude displacements allow investigations of anharmonic behavior. Currently, the most generally used method for evaluating phonon scattering based on MD simulation is to compute PDOS by obtaining the Fourier transform of each atom’s velocity autocorrelation function.

In this study, several aspects are addressed, such as the PDOS, the radial distribution function, the infrared spectrum and the effects of the interaction of different models of water with TNTs. These studies will provide information on the vibrational properties and structure of TNTs, as well as their interaction with the environment, aspects that are of great importance in various fields such as photocatalysis, electronics, solar energy, and composite materials. Understanding how energy is distributed in TNTs can help improve photocatalytic efficiency, conductivity and other electronic properties, solar energy conversion efficiency, and the mechanical and thermal properties of composite materials. Although numerous studies have been performed on the structural analysis of TNTs and nanosheets (TNS) in different crystalline phases, such as anatase and rutile, no comparative study has been performed using the radial distribution function (RDF). This comparative approach could shed light on how the geometry of the nanostructures and the nature of the phase affect the physical and chemical properties of TiO${}_{2}$. It could also help to establish relationships between the structural features and functional properties of TNTs and nanosheets. As for the theoretical approach, there are studies that extensively use the quantum approach based on DFT, which we also noted in our recent review [28]. A dearth of studies addressing the PDOS in TiO${}_{2}$ nanostructures using classical molecular dynamics (CMD) can be observed [29,30]. In particular, Ghuman et al. [30] analyzed amorphous and rutile nanoparticles to investigate the PDOS differences between them. However, to date, no recent studies have been found that apply CMD to investigate this property in TNTs and nanosheets. The same situation is repeated when investigating the optical properties of TNTs, as there are very few studies, and most of them are based on theoretical methods such as DFT [31,32]. In particular, a traditional CMD approach was used to analyze the adsorption structures of dimetridazole on anatase TiO${}_{2}$ (1 0 1) and (0 0 1) crystal surfaces using the LAMMPS [33] program. They also investigated RDF evaluation based on CMD for water droplets containing 1400, 1200 and 700 water molecules on Al${}_{2}$O${}_{3}$ and TiO${}_{2}$ surfaces [34]. They studied the phase transition from liquid to solid in water droplets on these surfaces, where there is crucial information about their stability and thermodynamic properties. Recently, the ReaxFF force field-based NVE–MD of the adsorption and degradation of metronidazole on the surfaces of TiO${}_{2}$ (101) and (001) was observed [35]. Similarly, ReaxFF simulations [36] were performed to calculate the volume difference between the amorphous and anatase phases of TiO${}_{2}$ in their crystalline phases. We emphasize here that one of our motives is to study how water behaves on anatase and rutile nanotubes and nanosheets. To our knowledge, no previous study has been performed on the CMD evaluation of PDOS and its influence in each dimension (x, y, z) and/or comparative evaluations of TNT and TNS. Similarly, one aspect that we highlight in this study is the soft-phonon effect, for which studies with graphene and SiO${}_{2}$ are taken as a reference, which, although they do not provide results that we can compare with our material, do offer us a guideline to understand the PDOS results. Since the phenomenon of the soft-phonon effect is classically unexplored in TNT and TNS. Using the CMD technique, we fill a gap in our understanding of the properties of TNT and TNS in their crystalline phases (anatase and rutile) and calculate their interaction and hydrophilic behavior with water.

## 2. Results and Discussion

### 2.1. Strutural Analysis in the Aspect of RDF

As shown in Figure 1a, a detailed comparison of the structural characteristics of anatase and rutile NSs and NTs at a temperature of 300 k and a pressure of 1 atmosphere was carried out. For these models, periodic conditions were applied in each and every direction. Examining the results, it should be noted that in almost all RDF curves, the peaks of the rutile nanostructures are higher and more dominant than the curves of the anatase nanostructures, both cases having in common that they have regular and very well-defined coordination shells. In the case of Ti-O bonding in both anatase and rutile NSs and NTs, it can be observed in Figure 1a,b that the RDF curves are similar, but two sharp peaks within the 3 Å cut-off radius stand out, which clearly correspond to the Ti-O bonding distances reported in experimental studies [37] for rutile (1.946 Å and 1.984 Å) and anatase (1.937 Å and 1.964 Å), noting that the rutile nanotube peak is slightly further to the right than the anatase nanotube peak. With this first observation, we can ensure that the equilibrium geometry of both the rutile and anatase nanotubes and layers has not lost its crystalline structure, as it is common for the peaks in amorphous structures to be smoothed [38,39,40,41,42]. In this case, this feature cannot be verified; on the contrary, they are very well defined. On the other hand, the silhouette of the RDF curve of the rutile nanolayer with respect to the O-O bond in Figure 1c is very different from the curve of the anatase nanolayer. Within the first 3 Å coordination shell, the rutile nanolayer has one large dominant peak, while the anatase nanolayer has three peaks of intermediate height. With respect to the NTs in Figure 1d, the RDF curves are very similar, with the rutile peaks only very slightly predominating over the anatase peaks. Similarly, one can see that in the Ti-Ti bond, from the second coordination shell onwards, the anatase nanosheet peaks are more dominant than the rutile NS. In contrast, this situation is not found in the NTs in Figure 1f, as upon inspection it is found that there are only three dominant rutile NT peaks and, in contrast, for anatase NT there are at least seven regular peaks. Another remarkable aspect is that in the RDF curve in the Ti-O bond, the anatase nanolayer and the anatase tube decrease their likelihood factor by 25% despite having the same number of atoms in the first shell coordination; we cannot verify this property in the rutile nanolayer and the rutile nanotube because the crystal structure is different and contains fewer atoms.

### 2.2. Phonon Density of States (PDOS) Analysis

Results were generated from CMD trajectories of nanotube systems with a radius of 10 Å and anatase and rutile nanosheets with 684 and 576 atoms, respectively. First, we confirmed that both TNTs and NSs of both TiO${}_{2}$ phases exhibit entropy dominated by low-frequency modes, especially below 12 THz (about 400 cm${}^{-1}$), which show a significant degree of inherent anharmonicity [43]. Nevertheless, the densities of states (PDOS) differ in each case because of their unique ability to represent the vibrational states of phonons. It is obvious that the nanosheets of both phases exhibit an even higher degree of anharmonicity in the PDOS profile than the nanotubes of these materials, which is why the phonon smoothing effect occurs, referring to the decrease in frequency and intensity. Similarly, to ensure our observation, it is crucial to note that recent research on carbon nanotubes [44] has shown that low-frequency phonon modes are particularly sensitive to tube diameters, and the vibrational spectrum of a carbon nanotube is close to that of graphene in large diameters. In parallel to this work, in effect, the nanosheets of both phases present a greater degree of anharmonicity in the PDOS than the nanotubes of the same materials. This can result in the phonon smoothing effect, which refers to the lowering of the frequency and dispersion of vibrational modes. The dispersion character of the vibrational modes is related to a higher probability of low-frequency modes at the surface, which is naturally caused by the reduction in confinement and the greater freedom of movement of the surface atoms. Similar results were recently obtained by Nil et al. and others [45,46,47]. Their study confirms that the normal mode spectrum for the 5-layer graphene surface layer shows significant differences from the intermediate layer with a higher proportion of low-frequency modes (6 THz), noting that, irrelevant to their material of interest, this may be an analogue to support our observations in TNTs and the NSs PDOS profile. Surface phonons are smoother than bulk phonons because surface atoms have a lower coordination number than bulk atoms. As a result, the frequencies of phonons [43] are generally lower. The results along the different directions are equivalent for 3D isotropic systems and can be averaged, but for 2D materials such as graphene, it is useful to consider the in-plane and out-of-plane components separately (the x, y and z directions) [48]. Of course, graphene and TiO${}_{2}$ are different materials, but it can be deduced that the phonon smoothing effect occurs in nanotube (1D) and nanosheet (2D) nanostructures in geometrically similar situations. Meanwhile, it is interesting to note that the anharmonic fluctuations of PDOS in TiO${}_{2}$ nanostructures can facilitate charge transfer and electron-hole pair generation, which is essential for photocatalytic activity. Recent studies support the idea of higher photocatalytic activity in TiO${}_{2}$ nanotubes and fragmented nanotubes compared to TiO${}_{2}$ nanoparticles [49]. Considering that nanosheets are also known as fragmented nanotubes, we establish a correlation between these theoretical–experimental findings and our PDOS as shown in Figure 2.

On the other hand, considering that the number of atoms of the studied nanostructured systems are different, but not the geometric nature, and although it was not our intention to observe the phase transition, a separate simulation of each MD nanosheet and nanotube based on an investigation of PDOS with the use of our implicit technique showed that in the transition from the 2D nanosheet to the 1D nanotube, both phases exhibit a higher maximum intensity for the entire spectral range of 0–1200 cm${}^{-1}$. In Figure 2, it is observed that both the anatase phase and the rutile phase of the nanotubes present higher intensities compared to the corresponding nanosheets. This result supports the idea that the high peaks and the number of state densities in the nanotube compared to the nanosheet indicate that the photocatalytic activity of TiO${}_{2}$ nanotubes is higher than that of TiO${}_{2}$ nanotubes, fragmented (also known as nanosheets). Besides that, as far as we know, there is no record of previous studies on the individual dimensional contribution to the density of states of these systems since we can see that there is a strong contribution of X and Y dimensions with the total PDOS, and the Z dimension has the lowest correlation with total system PDOS. A possible reason for this could be that the (001) orientation of the atoms is aligned in the X and Y dimensions to show the conformational modes rather than in the Z dimension for the geometric nature of the respective nanosystems. In particular, the anatase nanosheet in the x-dimensional PDOS profile showed no difference in the high-frequency doublet peak of 811 cm${}^{-1}$ and around 936 cm${}^{-1}$ from the total PDOS, while the two peaks were absent in their y- and z-dimensional counterparts. This is due to the fact that the x direction is parallel to the Ti-O bond in the nanosheet.

In addition, it is appropriate to mention the previous results of three-dimensional (3D) studies of metallic carbon [50], in which PDOS projections were made in three Cartesian directions to analyze phonon frequencies. In these studies, the high-frequency modes associated with the stretching of double bonds were observed only in the x-direction, while the y- and z-directions did not exhibit high-frequency peaks, see Figure 2. Although these results are not directly relevant to the properties of the material we studied, the observed phenomena could be comparable and contribute to our understanding of TiO${}_{2}$. In our study, we ensured that the configuration of the opposite mode of motion of oxygen atoms in adjacent O-Ti-O bonds in the anatase nanosheets is parallel to the x-direction. However, this is not the case in the y and z directions. This consideration is important when considering the geometric counterparts of the 2D nanosheets. We may recall that these low-frequency modes are usually attributed to a torsion and bending of both the anatase and rutile crystal assembly and their 001 projection to form their nanotube and laminas, at a temperature of 300 K. All atoms in the x and y, z dimensions participated in different ways as well as individually for their contribution to PDOS in the very low frequencies. Considering the range of 0–70 cm${}^{-1}$ (corresponding to 2 THz), individual peaks were observed for each direction. In particular, there is a large gap between each adjacent mode: in the case of the peak of the z-dimension of 23.068 cm${}^{-1}$, 47 and 70 cm${}^{-1}$ are seen; in the case of the peak of the x-dimension, 31 and 62 cm${}^{-1}$ are seen; and in the case of the peak of the y dimension, 39 cm${}^{-1}$ is seen. For the total PDOS, the combination of all modes is shown, with the high intensity of the very lowest mode being measured at 23.068 cm${}^{-1}$.

We also establish the understanding of our results with the existing recent experimental and theoretical findings on other similar materials, because although the vast majority of work on metal oxides refers to quantum DFT calculations, it is interesting to compare our results with those. Before prefacing these quantum observations, we noted a recent CMD study on a SiO${}_{2}$ polymorph, in which a fourth-order gear algorithm with a time step of 1.0 fs to integrate the isokinetic Gaussian equations of motion at 300 K and a pressure of 2 GPa revealed that the power spectra of silicon and oxygen atoms differ from each other, and the low-lying acoustic modes propagate at higher frequencies in the orthorhombic structure of SiO${}_{2}$ than in the hexagonal structure [45]. Although there has not been a direct comparison of the vibrational states of the two distinct phases of TiO${}_{2}$ in a classical picture, this distinctive PDOS profile observed by [45] in the hexagonal and orthorhombic phases of SiO${}_{2}$, studied using classical simulation methods, may be a better analog for the findings of the distinction in our PDOS of the nanosheet and nanotube in TiO${}_{2}$, (as shown in their Figure 6). As previously stated, no previous research on the comparative analysis of the TNT has been undertaken to evaluate our findings. Furthermore, in a recent study, another nonmetallic oxide material, graphene, and its various polymeric compositions was investigated by [46], in which nonequilibrium MD calculations on graphene and single-layer biphenyl nanoribbons produced an external strain along the x-direction to better understand the 1D heat transfer capabilities of these materials. Their results of the PDOS (as shown in their Figure 5c) in the x-direction and the overall PDOS showed that an increase in the strain can cause a red shift in both graphene and biphenyls, which was necessary for understanding the 1D heat transfer properties along the x-direction of nanoribbons of both biphenyls and graphene monolayers [46].

Hence,these observations could be indirectly correlated to our findings; for example, their biphenylene and graphene monolayers are structurally different but share a material class of 2D, in which there is an abrupt change in their harmonic vibrational modes from 0 to 2 THZ as shown in their Figure 5c. They show in their Figure 5c that low-frequency modes of less than 2 THz are less dense in the x-dimensional case than their overall atomic modes; the same phenomenon can be seen in our individual dimensional case (shown in Figure 3) for both the 1D nanotube and 2D nanosheet of two different TiO${}_{2}$ phases. Although graphene is not part of this study, reviewing a study of a 5-layer sheet [47] shows that the top layer has more conformational modes than the middle layer, indicating that surface phonons are smoother than bulk phonons because surface atoms have a lower coordination number than bulk atoms. Comparatively, using this idea as an analysis strategy, we found that anatase and rutile have more conformational modes in their nanosheet structure than the nanotube. This could imply that the surface phonons in the unwrapped nanosheet are higher than in the wrapped nanotube. Thus, it indicates that nanosheet phonons are smoother than nanotube phonons because the surface atoms of the nanosheet have a lower coordination number than the nanotube atoms. This results in a general decrease in the nanotube phonon frequencies as illustrated in our Figure 3. This peculiar behavior of conformational modes at low frequencies drew our attention to the multilayer TiO${}_{2}$ nanosheet and its impact on the phonon softening effect. As a summary, these recent results [45,46,47] are well supported and documented for our nanosheet and nanotubes of TiO${}_{2}$, especially the surface phonon behaviors and their contribution to low-frequency modes of TiO${}_{2}$, which meansthat the effect of phonon softening correlates with a higher probability of low-frequency modes at the surface of TiO${}_{2}$ nanostructures; this effect is one of the crucial phenomena for comprehending the various sizes and thicknesses of TiO${}_{2}$ nanostructures.

We can only speculate on how some of our data might be interpreted after looking at Figure 3.

The discovery that even the peak frequencies of anharmonic modes generally correspond to the low-frequency modes with which they are most closely associated (see Figure 3) has already been mentioned. Low-frequency modes indicate the globular movement of the all-atom system. We try to observe the harmonic behavior of rutile nanosheets in the very low frequency range of 0–80 cm${}^{-1}$, which corresponds approximately to a value of 2 THz, when one part or group of molecules oscillates and the other part of the system is at rest, so this participation can also occur in certain groups of atoms in certain dimensions as we see that in the case of the rutile nanosheet, periodic oscillations were observed. The first frequency was close to 20 cm${}^{-1}$ in the z dimension, the second was close to 30 cm${}^{-1}$ in the x dimension, and the third was close to 40 cm${}^{-1}$ in the y dimension. The situation is similar for the other conformational modes, although we must emphasize that these frequencies may not be as precise as a Hessian-based harmonic approach to determine the best possible lowest natural frequencies. Despite the fact that these modes are our most consistent and possibly comparably lowest frequencies, an important significance of this plot is that there were six peaks for the entire PDOS case, and likewise, each dimension shared two peaks, indicating the softness of the surface phonon oscillations.

In Figure 4, where the PDOS of an anatase nanotube at 300 K temperature is examined by each type of atom, we find another pertinent conclusion. The PDOS of the oxygen atoms in this graph is quite similar to the total PDOS curve of the nanotube, while the PDOS of the titanium atoms shows a striking contrast with bigger peaks at low frequencies (0–200 cm${}^{-1}$) and lesser peaks at higher frequencies (700–800 cm${}^{-1}$). This demonstrates that the anatase nanotube’s number of oxygen atoms has a significantly higher impact on the power spectrum generated by the entire system than the titanium atoms do. In general, the PDOS of all the systems show well distinct peaks in its PDOS spectrum (see Figure 2 and Figure 3).

Curiously, in previous studies using another type of material, such as rutile-type GeO${}_{2}$, a PDOS analysis was carried out using atomic contributions, highlighting that the movements of germanium and oxygen are involved for up to 620 cm${}^{-1}$, while only oxygen movements strongly dominate the two doublets around 900 cm${}^{-1}$. In contrast, the PDOS of the rutile type GeO${}_{2}$ [51] has a more continuous profile between 0 and 900 cm${}^{-1}$ due to the stronger dispersion of its optical branches (as shown in their Figure 2). Germanium motions are now restricted in the 100–500 cm${}^{-1}$ range, while oxygen motions strongly dominate the modes above 500 cm${}^{-1}$ but also contribute around 350 cm${}^{-1}$.

On the other hand, in considering the atomic contribution, four typical vibrational modes from a Raman spectral feature were observed for TiO${}_{2}$ around 145 cm${}^{-1}$ (B1g), 445 cm${}^{-1}$ (Eg), 610 cm${}^{-1}$(A1g), and 240 cm${}^{-1}$ for the second-order effect (SOE) [52].The qualitative quality of the trend is consistent and comparable for prediction purposes, even though the vibrational power spectrum should not exactly match the experimental IR/Raman spectra. We focused only on the NT of the anatase phase, as it can be seen in Figure 4 that at frequencies below 150 cm${}^{-1}$, a strong influence of the intensity of Ti atoms PDOS is observed. It prevails that the Ti intensity is higher than that of the O atoms. This could support the recent results of the experimental Raman spectroscopy of samples of undoped anatase nanopowder annealed at 400 °C [53], which have contributed to the dominant B1g at 144 cm${}^{-1}$, which is associated with the asymmetric bending of the O-Ti-O bonds.

We therefore estimate that our spectral features at very low frequencies could detect the B1g dominant peak for the PDOS TiO${}_{2}$ NT that relates to the Ti-O bending motion. Also, we could observe that the Eg mode around 445 cm${}^{-1}$ is characterized by the asymmetric bending of the O-Ti-O bonds caused by the opposite movement of the O atoms across the O-Ti-O bond. The A1g mode around 610 cm${}^{-1}$ is characterized by the symmetric stretching of the O-Ti-O bonds, caused by the opposite movement of the O atoms in the adjacent O-Ti-O bonds. Moreover, Ti spectral features are dominant from 0–570 cm${}^{-1}$, and in the mid-frequency range of 570–740 cm${}^{-1}$, both Ti and O contribute equally to the total PDOS. At the highest frequencies of 740–900 cm${}^{-1}$, Ti is probably less dominant than the oxygen atoms.

All conformational modes display a different vibrational order in the region of 0–9 THz, regardless of temperature fluctuation. The anatase phase exhibits horizontal as well as dense vibrational states, indicating that the surface phonon behavior and the associated softening effect are different. This implies that the phonon properties of TiO${}_{2}$ nanosheets that are structurally (001 projection) and geometrically similar are distinct. Rutile nanosheets exhibit increasing mode order, while anatase nanosheets exhibit relatively dense peaks. If the temperature effect is also taken into account, we can see that even if the intensity is unchanged, it is clear that the anharmonicity significantly affects the phonon modes. In particular, a red shift, as would be expected from an increase in temperature, is visible for 400, 500, and 600 K in the range of 200 cm${}^{-1}$ to 800 cm${}^{-1}$, as shown in Figure 5.

In accordance with previous research [54,55,56,57,58,59,60], it has been shown that the average total energy of TiO${}_{2}$ nanotubes is correlated with their diameter. This was corroborated in our calculations performed on tubes of different sizes (10 Å, 30 Å, 40 Å, 50 Å). Figure 6 shows that the total energy of TiO${}_{2}$ TNTs is inversely proportional to their diameter, converging toward a maximum diameter of around 50 Å. This is because the diameter of the tube describes its curvature, and more curved TNTs may have atoms with a more distorted geometry, requiring more energy to keep the structure stable. Additionally, steeper bends in a nanotube can generate internal stresses, which also increases the total energy of the structure.

As can be seen in Figure 7, as the radius of the TNTs and thus the number of atoms increases, the PDOS also increases significantly at all frequencies. At low frequencies, we notice that in the case of 3 THz, compared to PDOS of the 10 Å nanotube, the nanotube with 30 Å radius increases by 7.19%, the nanotube with 40 Å radius increases by 11.13%, and the NT with 50 Å radius increases by 16.39%. Similarly, we find that at the frequency of 5 THz, the tubes with the largest radius increase by at least 3% compared to the PDOS of the tube with the smallest radius. Similarly, at the frequency of 10 THz, it was calculated that the PDOS increase in the 40 Å radio nanotube is 14.10%, 16.87% for the 50 Å radio nanotube, and 18.75% for the 30 Å tube NT.

### 2.3. Infrared Spectrum (IR) Analysis

The anatase structure is attributed to the Raman bands in NT -TiO${}_{2}$ at 127, 153, 201, 253, 414, 473, 519, and 640 cm${}^{-1}$ [61], similar to the experimental observation. Here, we note that these results agree well with our dipole autocorrelated power spectrum, in which it identifies a similar spectral profile and corresponding peaks of the experimental Raman (Figure 8), particularly for the spectral region of 400–800 cm${}^{-1}$. Moreover, 153, 201 and 253 cm${}^{-1}$ are missing in our dynamical spectra; this is because the anharmonicity of the vibration reduces the intensity of these particular modes, and we also find that there should be a harmonic coupling of these modes of 201 cm${}^{-1}$ and 253 cm${}^{-1}$, which are harmonically coupled, which is highlighted in the simulated spectra almost at 300 cm${}^{-1}$. More on this, we could see that a tiny part of the peak is seen at 153 cm${}^{-1}$ (Figure 8), which agrees well with the experimental observation. Moreover, our calculated spectra ensure that all possible frequencies of TiO${}_{2}$ NT are visible; in order to elucidate each peak of IR frequency in CMD, it might be necessary to treat the modified potential to capture the low frequencies in the range of 0 to 9 THz, which are associated with the localized modes of TiO${}_{2}$. In addition, our study could lead to an understanding of the influences of temperature on the absorption factors. Recent Raman spectroscopy results for the conversion of amorphous to anatase NT were heat treated at 450 °C [62]. Whereas the measurement factors for anatase, the peaks associated with 150 cm${}^{-1}$, 350 cm${}^{-1}$, 470 cm${}^{-1}$, and 580 cm${}^{-1}$, were seen, the amorphous nanotube did not show any of these peaks in their observation. In addition, we note that the inset in Figure 8 shows that the first low frequency corresponding to the experimental Raman of 127 cm${}^{-1}$ is seen at 121 cm${}^{-1}$ for the temperatures of 300 and 400 K; interestingly, the peak shifts to 128 cm${}^{-1}$ at higher temperatures (500 K and 600 K), although the low-frequency modes corresponding to 127 cm${}^{-1}$ are definitely present, and also the intensity of this peak decreases compared to those at 300 and 400 K, suggesting that there is a blue shift due to the temperature increase.

A valuable finding that can be evidenced in Figure 9 is that by immersing an anatase nanotube in three flexible-type water models, the infrared spectrum significantly decreases its intensity, and the peaks of the representative frequencies of the nanotube that in a vacuum are notaries in the water models are smoothed out. This may be largely due to the influence of a large number of water molecules that interact with the anatase nanotube, and as shown in Figure 10a, according to each type of water model, there is a greater number of molecules confined to the tube.

Over the past 4 decades, 30 water models have been developed, and it remains a challenge to find a single water model that accurately reproduces all experimental properties of water simultaneously. In our case, we considered (i) CVFF, an equivalent of the flexible SPC water model. Studies have shown increased free energies of hydration [63,64] and their internal geometry corresponding to the experimental gas phase structure. (ii) TIP3pFW type 3 points to a flexible version of the rigid model of TIP3P, and (iii) similar to CVFF, another flexible model of the class II COMPASS force fields used, they contain cross terms to better reproduce experimental vibrational frequencies and are primarily intended for polymers and material applications. These three flexible models were chosen because they are sufficient to completely comprehend the interaction of nanotubes and water. As seen in Figure 10a, the amount of water molecules in the various flexible water models produces dramatically diverse results. The progress of these water models in adsorbing on the surface of NT can also be visualized in the Appendix A because the CVFF model has a larger number of water molecules confined to a distance of 3 Å, while the COMPASS model has the smallest number of molecules in this first layer. Although the SPC/E water model was not considered in our work, it is worth mentioning that the interaction between water and anatase (101) TiNTs with a diameter of ∼1 nm was studied using the extended simple point charge (SPC/E) model [63], suggesting that for water inside tubes, the average number of hydrogen bonds ⟨nHB⟩ follows this order, TiNT (480 atoms) < carbon modified TiNT (1082 atoms) < carbon nanotube (480 atoms), while the relaxation times follow the opposite due to fewer dangling -OH bonds of water molecules in the modified TNT. On the other hand, we were curious about the influence of these three different water models on a larger nanotube system (rutile and anatase NT have 576 and 864 atoms, respectively, as shown in Figure 11) compared to the pure versus doped TNTs. Figure 10 and the accompanying visualization video of the MD trajectory (Appendix A) suggest that the adsorption of a larger number of water molecules onto a TNT in the CVFF model is due to a variety of factors related to the quality and properties of the water model used. The CVFF model has an undulating profile, whereas the other two models have a non-undulating, quasi-stable shape. On the one hand, the CVFF water model clearly contains features that support a stronger interaction with the TNT. Thus, the shape and surface charge of the TNT can both influence water molecule adsorption. As a result, the CVFF water model is characterized as better describing the electrostatic or van der Waals interactions with the nanotube surface.

## 3. Materials and Methods

### 3.1. Creating the Nanotubes

To create several models of TNTs with different geometric parameters (i.e., internal radius and wall thickness) and microstructures (rutile, anatase, or amorphous), we wrap the sheet created by applying the geometric principles in our in-built Python code, which implements the spatial transformation method to convert the sheets into nanotubes. The anatase nanotube’s axis is aligned with the [001] direction, but it is vital to note that the structure of the nanotube was analyzed as if it were an infinite structure in the z dimension. The structures included for this simulation are depicted in Figure 11.

In terms of inter-atomic potentials, the Matsui and Akaogi potentials, as well as the Lennard–Jones reparametrization of the Buckingham potential, are commonly used empirical models that characterize titanium dioxide nanostructure interactions. The Born–Huggins–Meyer potential (BHM) is a derivative of the Matsui and Akaogi potential (MA), which is based on a mixture of pairwise and three-body potentials [65]. This is a hybrid model that combines, on the one hand, the Buckingham potential, which represents the short-range attractive and repulsive interactions between atoms, and, on the other, the Coulomb term, which describes the electrostatic interactions between charged atoms. The expression represents the potential MA [66,67]:(1)$$U\left(r_{ij}\right)=A_{ij}exp\left(\frac{-r_{ij}}{\rho_{ij}}\right)-\frac{C_{ij}}{r_{ij}^{6}}+\frac{q_{i}q_{j}}{r_{ij}}$$
where *U*(*r*${}_{ij}$) is the potential energy as a function of the interatomic distance *r*${}_{ij}$, and their charges are $q_{i}$ and $q_{j}$. The parameters $A_{ij}$, ρ${}_{ij}$ and $C_{ij}$ are intended for anion–anion, anion–cation, and cation–cation interactions. Cation–cation interactions are described solely by their electrostatic nature. The exponential part of the Buckingham potential is responsible for explaining the repulsion that occurs between the layers of linked electrons, this being a consequence of the Pauli exclusion principle. On the other hand, the negative power term describes both an induced dipole attraction and a van der Waals interaction, which manifests itself when there is some asymmetry in the electron density distribution, and represents the electron density. The parameters for the Buckingham potential interaction are summarized in Table 1.

In the Lennard–Jones re-parametrization of the Buckingham potential [66,68], the parameters of the original potential are adjusted to fit a particular system or material. This potential allows greater flexibility in fitting the model to specific conditions and improves its accuracy. In addition, it can be implemented in the LAMMPS package without the need to develop a new potential from scratch. It is expressed by the following equation:(2)$$U\left(r_{ij}\right)=\epsilon_{ij}\left[{\left(\frac{\sigma_{ij}}{r_{ij}}\right)}^{12}-2{\left(\frac{\sigma_{ij}}{r_{ij}}\right)}^{6}\right]+\frac{q_{i}q_{j}}{r_{ij}}$$
where $U(r_{ij}$) represents the potential energy as a function of the equilibrium inter-atomic distance $r_{ij}$, and their charges are $q_{i}$ and $q_{j}$. ϵ is the attractive energy parameter that reflects the energy depth between the particles, which defines the size of the attractive force between them, whereas σ represents the distance at which the potential achieves its minimal value. It is the strongest point of attraction between the particles. Unfortunately, because the Lennard–Jones potential assumes an isotropic and spherical interaction [68,69,70], it does not take into account crystal structures, making it difficult to characterize the unique equilibrium forces and distances in anatase and rutile crystals. This approach may not be appropriate for TNTs with a specific elongated and geometric structure [71]. The potential of Matsui and Akaogi (MA) is better suited to describe the interactions in complex molecular systems, such as TNT [67,72,73,74]. Indeed, the potential (MA) allows a better description of non-spherical interactions and can more accurately reflect the structural properties of the system. Studies have confirmed that the potential MA gives better results for TNT compared to previous experimental or theoretical data [66,75]. For this reason, this empirical model was used in this study. The Coulomb interactions were estimated in our simulations using the Ewald summation method, which is designed to quantify long-range electrostatic interactions in periodic systems. Splitting the potential into short-range components, periodic grids, and offsets results in an accurate and efficient approximation for calculating these interactions, which is very valuable in MD simulations of charged systems, such as TNTs [76].

The following MD methodology was used to analyze the post-processing of the trajectories using the LAMMPS simulation package [77]. For the relaxation phase of the nanotubes and nanosheets, the NPT ensemble was utilized for 450 ps at a temperature of 300 k and a pressure of 1.013 bar, then the NVT ensemble for another 450 ps with the same thermodynamic values, and finally the NVE ensemble for 100 ps with a time step of 0.001 ps. During the production phase, the NVT ensemble was employed with different simulation times. In the production phase, the NVT ensemble was used with different simulation periods. In the case of PDOS, it was simulated for 2 ns, and only the velocity trajectories of the last 200 ps were stored to ensure the stability of the system. For the calculation of the infrared spectrum, on the other hand, it was run for 2 ns and stored every 2 ps. To produce PDOS and IR signatures, the respective systems were simulated for 2 ns, in which the velocities and positions of the last 200 ps with every 2 femto seconds were stored.

### 3.2. Radial Distribution Function (RDF)

RDF is defined as the probability of particle presence calculated assuming spherical layers around each particle with different radii, which allows to describe the variation in atomic density for which the number of atoms around a given particle in each of these coordination layers is counted [78]. The RDF, represented by the symbol g(r), as defined by Hansen and McDonald [79], is calculated using Equation (Equation 2): (3)$$x_{\alpha }x_{\beta }\rho g_{\alpha \beta }\left(r\right)=\frac{1}{N}\langle \sum_{i=1}^{N_{\alpha }}\sum_{j=1}^{N_{\beta }}\delta (r+r_{i}+r_{j}\rangle$$

### 3.3. Phonon Density of States (PDOS)

One of the most essential aspects of a solid state system is its phonon density of states, which influences the optical, mechanical, electrical and thermodynamical properties. Given that the power spectrum of the autocorrelation of the velocity is the PDOS in the harmonic approximation, we estimated the PDOS of rutile and anatase TiO${}_{2}$ 001 nanosheets and nanotubes, which is defined as follows [48]: (4)$$PDOS\left(\omega \right)=\frac{1}{3NK_{B}T}\int_{0}^{\infty }\frac{\langle \vec{v_{j}}\left(0\right)\cdot \vec{v_{j}}\left(t\right)\rangle }{\langle \vec{v_{j}}\left(0\right)\cdot \vec{v_{j}}\left(0\right)\rangle }e^{i\omega t}dt$$
where 〈$\vec{v_{j}}\left(0\right)$·$\vec{v_{j}}\left(t\right)$〉 is the velocity autocorrelation function (VACF), and, on the other hand, ω demonstrates the angular frequency, *N* is defined as the number of atoms, $K_{B}$ represents the Boltzmann constant, and *T* constitutes the absolute temperature. In the calculations of the total PDOS, *C*(*t*) is obtained by [80]
(5)$$C\left(t\right)=\langle \sum_{j=1}^{N}\vec{v_{j}}\left(0\right)\cdot \vec{v_{j}}\left(t\right)\rangle$$
$\vec{v_{j}}\left(t\right)$ represents the velocity of atom *j*, and describes the ensemble average. We then calculated the entire velocity’s autocorrelation. $v_{x}$, $v_{y}$ and $v_{z}$ were considered for this purpose, and calculations were made using only the velocity of each axis, i.e., for PDOS in x, only the velocity in the x-axis was considered. This technique was repeated for PDOS in y and z. PDOS was calculated using a function for groups of atoms (LAMMPS), which extracted the velocities for each type of atom (oxygen and titanium individually). The computation is performed by randomly assigning velocities to the system’s atoms and seeing how they change over time. A high number of initializations are often employed, with the results averaged. To assess the numerical Fourier transform across a finite interval, a Gaussian approximation is applied, resulting in a Gaussian stretching of the resultant spectrum.

### 3.4. Infrared Spectrum (IR)

Fourier transformation was used to obtain the infrared (IR) absorption spectra for the nanostructures from the system dipole moment ACF [81,82,83]: (6)$$I\left(\omega \right)\propto \int_{0}^{\infty }\langle M(t)\cdot M(0)\rangle cos\left(\omega t\right)dt$$
where I(ω), M(t), and are the system’s total dipole moment at time *t* and the vibration frequency, respectively. The vector sum of each individual molecular dipole moment of the nanostructures in the simulation box is derived as M(t) [81,82]: (7)$$M\left(t\right)=\sum_{i=1}^{n}\mu \left(t\right)$$
where μ(t) is the dipole moment vector of the *i*-th atom at time *t* and *n* denotes the number of atoms in the simulation box. The formula for the dipole moment’s ACF is the following [84]: (8)$$\langle M\left(t\right)\cdot M\left(0\right)\rangle =\langle \sum_{i=1}^{n}q_{i}r_{i}\left(t\right)\cdot \sum_{i=1}^{n}q_{i}r_{i}\left(0\right)\rangle$$
where *r*${}_{i}$(*t*) is the position vector of the *i*-th atom at time *t*, $q_{i}$ is the fixed electric charge of the *i*-th atom, and *n* is the total number of atoms in the system.

## 4. Conclusions

RDF analysis revealed TNT/NS simulations that correlate with experimental data and distinguish between the anatase and rutile phases in TiO${}_{2}$. The distinction of the anatase and rutile phases was demonstrated by the lack of previous studies that compared these two phases with 2D nanosheets and 1D TNTs of TiO${}_{2}$ using standard MD. Additionally, their RDF of particular atomic moieties, such as O-O, Ti-O, and Ti-Ti, showed the exclusivity in separating the anatase and rutile phases TiO${}_{2}$.

CMD validates the influence of phonon softening in TNT and TNS, with a higher chance of low-frequency modes seen on the surface of TiO${}_{2}$ nanosheets; moreover, this effect is one of the critical phenomena for understanding the varied sizes and thicknesses of TiO${}_{2}$ nanostructures. Spectral signatures based on velocity-autocorrelated power and dipole-autocorrelated IR spectra are also comparable to the experimental findings, explaining some of the discrepancies in the experimental data. Despite the fact that anharmonicity exists in molecular dynamics (CMD), the majority of experimental spectroscopic traces are observed in dipole-autocorrelated spectra. The following significant TiO${}_{2}$ experimental IR peaks were observed: 127, 153, 201, 253, 414, 473, 519, and 640 cm${}^{-1}$.

The stability of single-layer nanosheets and TiO${}_{2}$ TNTs was demonstrated and evaluated at temperatures ranging from 300 K to 600 K. Regarding solvent adsorption, specifically the adsorption behavior of the three water models, (a) Tip3P FW water model, (b) CVFF water model, and (c) COMPASS water model (refer to Figure 10), the COMPASS water model showed a stable profile on the surface of the TiO${}_{2}$ nanotube; on average, 120 water molecules are adsorbed on the surface of the nanotube, and this number varies for the other water models. Our preliminary observations suggest that metal oxide nanosurfaces and metal oxide TNTs can perform a variety of functionls, opening new opportunities for the development of optoelectronic nanodevices.

The temperature effects on vibration properties of the anatase TiO${}_{2}$ nanotube were clearly demsontrated through power spectral analysis. Indeed, the shift towards blue in the low-frequency peak (127 cm${}^{-1}$) indicates an increase in the energy of the vibrational frequency of that specific mode. Furthermore, the reduction in the intensity of this peak at higher temperatures indicates a decrease in the amplitude of the vibration related to said mode. In summary, it was established that the low-frequency vibrational oscillations in the anatase TiO${}_{2}$ nanotube are sensitive to the increase in temperature, resulting in a blue shift and a decrease in the intensity of the corresponding peak in the density of states. of phonons (PDOS).

Both the anatase phase and the rutile phase of TiO${}_{2}$ nanosheets exhibit different vibrational properties. The anatase phase exhibits horizontal and dense vibrational states, while the rutile nanosheets exhibit increasing mode order. This indicates that the phonon properties in these structures are different despite their structural and geometrical similarity in the 001 projection. The presence of anharmonicity significantly affects the phonon modes, and a redshift in the range of 200 cm${}^{-1}$ to 800 cm${}^{-1}$ can be observed for temperatures of 400, 500, and 600 K, respectively.

## Figures and Tables

**Figure 1 ijms-24-14878-f001:**
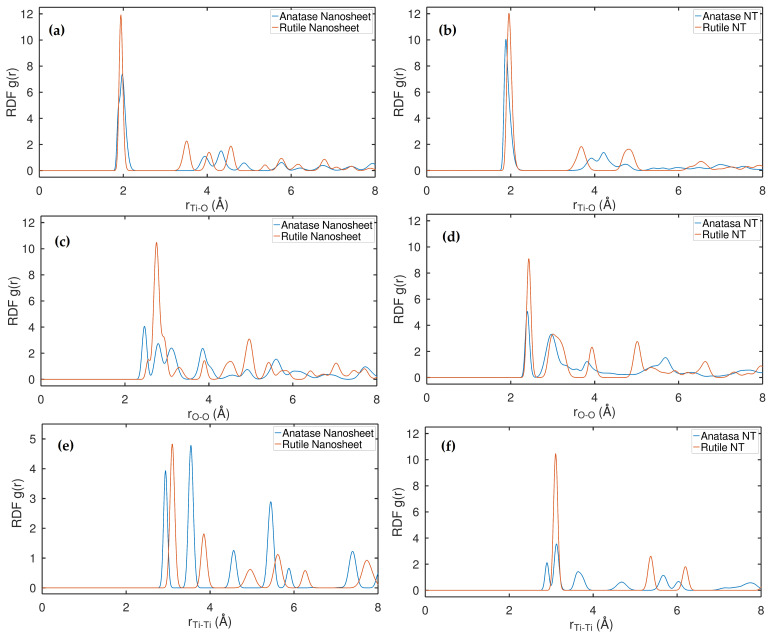
Radial distribution functions (RDFs). (**a**) Anatase nanosheet and rutile nanosheet pairs of Ti-O. (**b**) Anatase natube and rutile nanotube pairs of Ti-O. (**c**) Anatase nanosheet and rutile nanosheet pairs of g(r O-O). (**d**) Anatase nanotube and rutile nanotube pairs of O-O. (**e**) Anatase nanosheet and rutile nanosheet pairs of Ti-Ti. (**f**) Anatase nanotube and rutile nanotube pairs of Ti-Ti.

**Figure 2 ijms-24-14878-f002:**
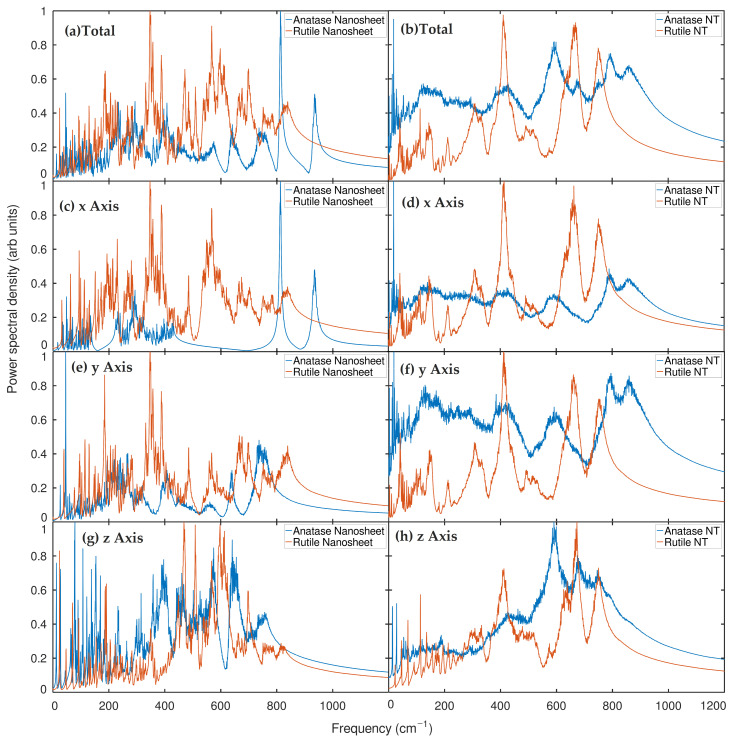
Phonon density of states of anatase and rutile nanostructures obtained from the Fourier transform of the velocity autocorrelation. (**a**) Total PDOS of anatase nanosheet and rutile nanosheet. (**b**) Total DOS of anatase nanotube and rutile nanotube; PDOS of anatase nanosheet and rutile nanosheet (**c**) x-axis (**e**) y-axis, (**g**) z-axis; PDOS of anatase nanotube and rutile nanotube (**d**) x-axis (**f**) y-axis, (**h**) z-axis.

**Figure 3 ijms-24-14878-f003:**
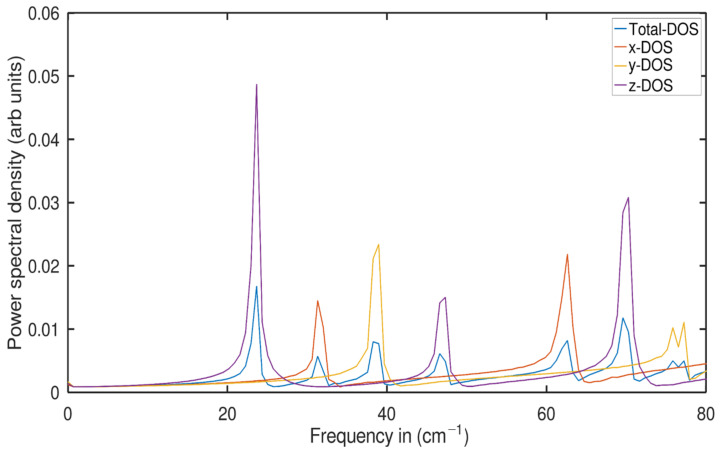
Power spectral density of each axis of a rutile nanosheet.

**Figure 4 ijms-24-14878-f004:**
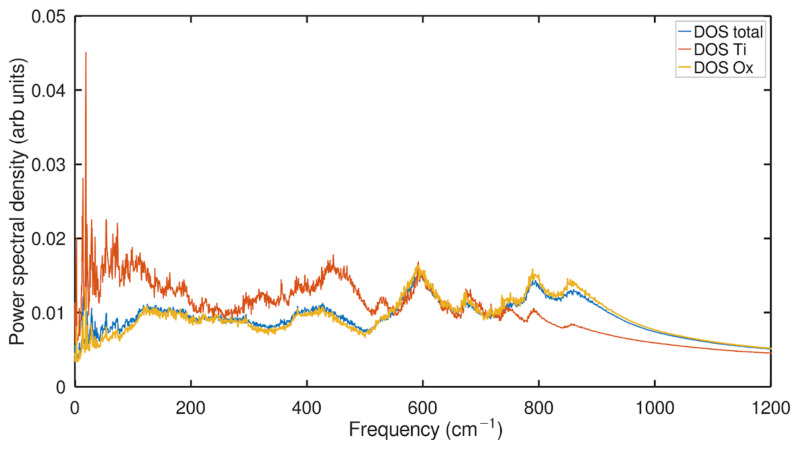
Anatase NTs density of states through calculations of the power spectrum for the whole TiO${}_{2}$ moiety (blue), Ti (red), and oxygen (orange) atoms.

**Figure 5 ijms-24-14878-f005:**
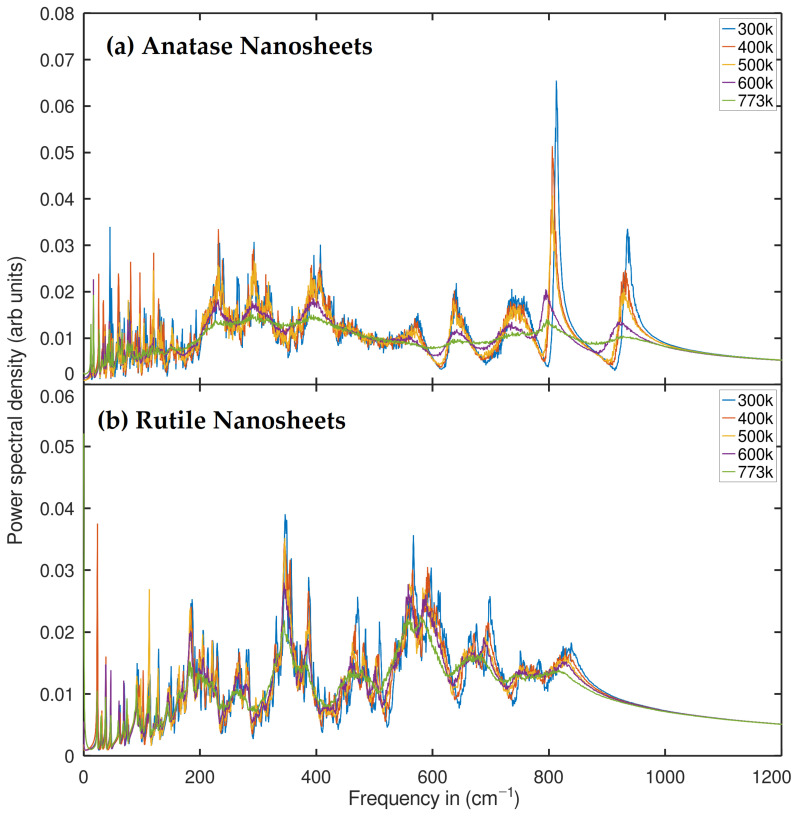
Phonon density of states of anatase and rutile nanosheets at different temperatures.

**Figure 6 ijms-24-14878-f006:**
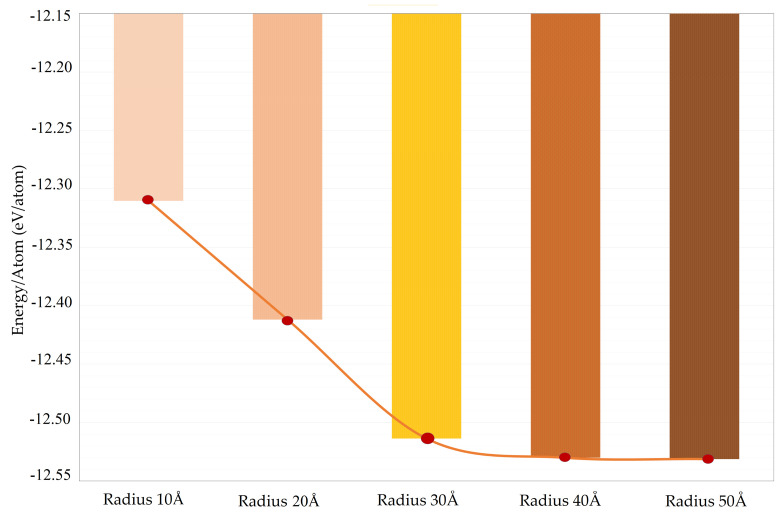
Average total energy of anatase nanotubes of varying radius.

**Figure 7 ijms-24-14878-f007:**
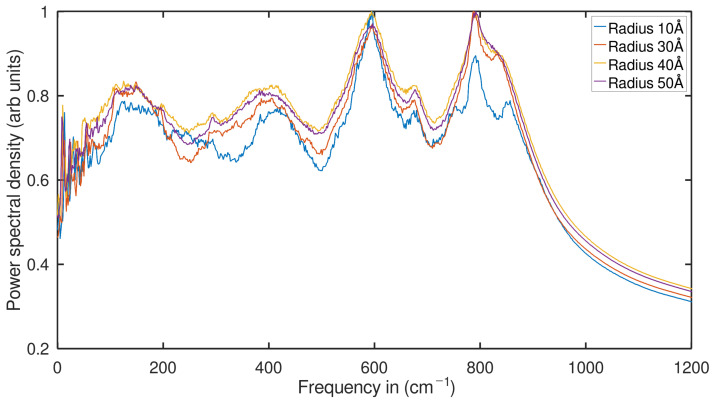
Analyze the anatase TNTs density of states using power spectrum analysis for tubes with various radii (normalized).

**Figure 8 ijms-24-14878-f008:**
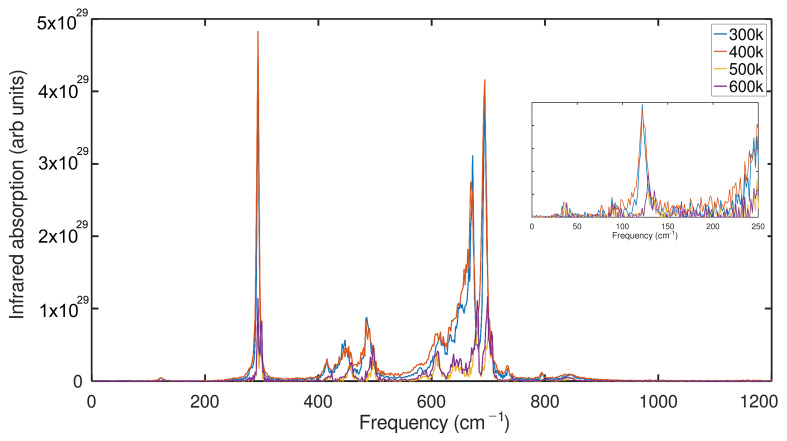
Computational infrared spectrum of Anatase NT with radius of 10 Å and length 31 Å in different temperatures.

**Figure 9 ijms-24-14878-f009:**
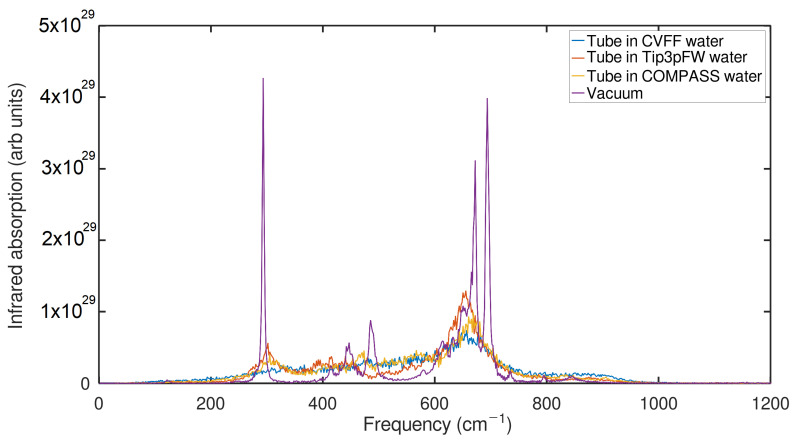
Spectrum of Anatasa NT in different water models and vacuum.

**Figure 10 ijms-24-14878-f010:**
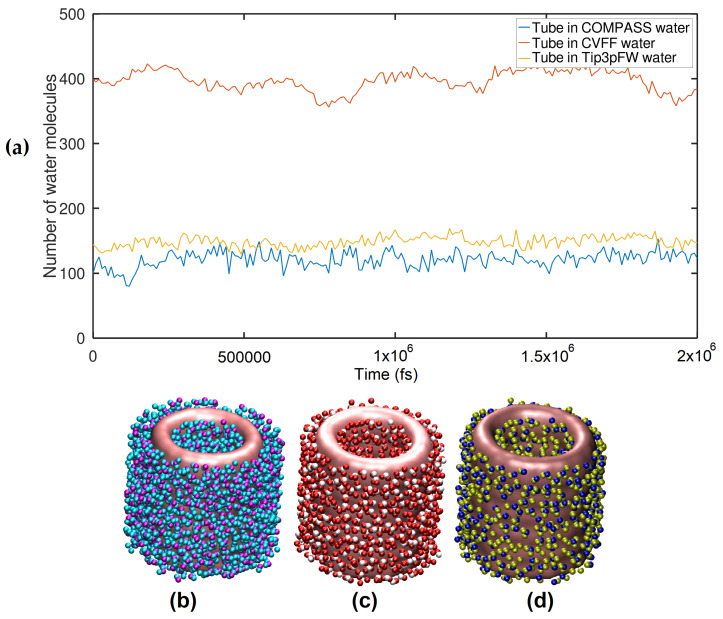
(**a**) Number of water molecules confined (3 Å) next to anatase NT in different water models, (**b**) water molecules confined (3 Å) next to anatase NT in CVFF water model, (**c**) water molecules confined (3 Å) next to anatase NT in Tip3P-FW water model, (**d**) water molecules confined (3 Å) next to anatase NT in COMPASS water model.

**Figure 11 ijms-24-14878-f011:**
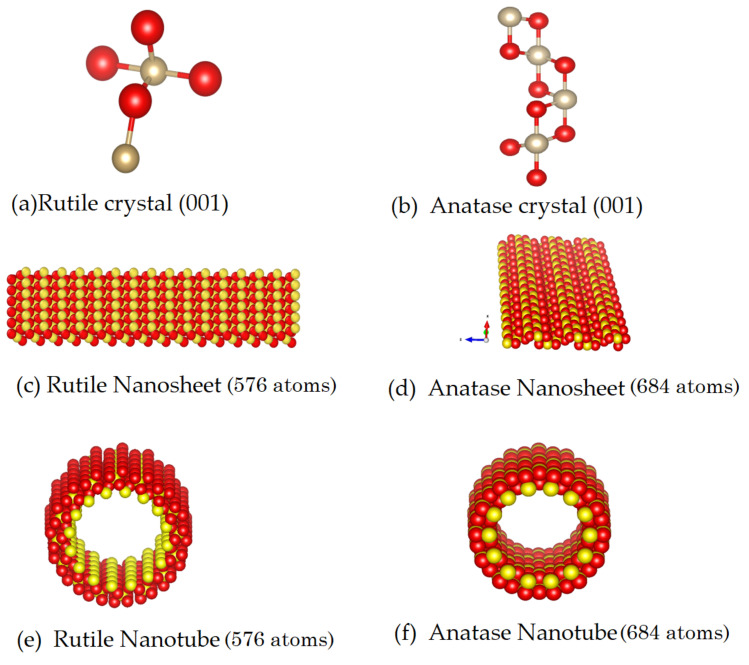
Initial structure of nanotube (**a**) in rutile phase, crystal in orientation 001 (**b**) in anatase phase, crystal in orientation 001. During the production dynamics of nanotube, at T = 300 K (**c**) in rutile phase, 18 nm × 73 nm, (**d**) in anatase phase, 29 nm × 72 nm. During the production dynamics of nanotube in water, at T = 300 K (**e**) in rutile phase, radius = 12 nm, length = 20 nm, at the (**f**) in anatase phase, radius = 12 nm, length = 20 nm.

**Table 1 ijms-24-14878-t001:** Coefficients of Buckingham potential proposed by Matsui and Akaogi.

Interaction	$A_{\mathrm{ij}}$ (Kcal/mol)	$\rho_{\mathrm{ij}}$ (Å)	$C_{\mathrm{ij}}$ (Kcal/mol)/Å
Ti-Ti	717,653.9571	0.154	120.9967
Ti-O	391,052.7442	0.194	290.3920
O-O	271,718.8311	0.234	696.9407

## Data Availability

Data generated or analyzed during this study are available from the corresponding author upon reasonable request.

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
