# Peer review of "Structural and Electromagnetic Signatures of Anatase and Rutile NTs and Sheets in Three Different Water Models under Different Temperature Conditions"

_ijms, 2023, doi:10.3390/ijms241914878_

Round 1

Reviewer 1 Report

The article present results of MD simulations of nanotubes formed by titanium dioxide molecules. The authors studied systems with different geometric parameters (nanotube diameters) and microstructures (rutile, anatase) in vacum and immersed in water. They also analyzed the temperature effects. The pairwise Matsui-Akagogi potential was used to model the interactions. Three water models lwere considered, Tip3P-FW, CVFF and COMPASS.

For various nanostructures, several quantities characterizing the system were calculated, namely radial distribution functions, the vibrational density of states, the infrared absorption spectra, and the total energy per atom. The authors compared the results obtained for anatase and rutile nanosheets.

The novelty is the determination of these characteristics for different structures of titanium dioxide and the detailed analysis of the results. The results are interesting and valuable. The articles is quite well written. Unfortunately, there are some minor typos or misleading formulations. For example,

- equation (3): in the integral exp(iωt)dt is missing,

- line 346, “states (VDOS) spectrum (see Fig. ??)” - a lack of figure number,

- line 446 titanium dioxide is written as TiO ,

- line 476 ,"As far as I’m aware, there aren’t many classical simulations available..." It should be rather "we are"?

- reference [29], wrong authors.

The manuscript should be carefully checked. I recommend the publication after minor revision.

Minor corrections would be advisable

Reviewer 2 Report

The main question addressed by the research is the structural and electromagnetic signatures of anatase and rutile TiO2 nanotubes under different conditions. The topic is not original, and I doubt as to the novelty of the obtained results, since there are so many computational studies these days concerning the TiO2 nanotubes that simulation of their power spectra, infrared spectra, water absorption, etc. seems to be somewhat not new. Take, for example, the works of Cao and co-workers (10.1021/acs.jced.6b00551), who investigated the molecular behavior of water on titanium dioxide nanotubes using molecular dynamics simulations. Reviewing such works is exactly what the authors ought to have done in the introductory section, but they haven’t. Thus, it is unclear what does their work add to the subject area compared to the other published material?

Above all, the manuscript is written badly (stylistically), inconsistent, poorly structured, hence, it is very hard to read. There are a lot of misprints and a great deal of redundant information that strays away the reader from the main narration.

If the authors totally rewrite their manuscript, make it clear and substantiate their goals as compared to the other works, I would recommend it for publication. But for now, a major revision is required.

Specific improvements:

1.      Important references are absent:

a.       W. Cao, L. Lu, L. Huang, Y. Dong, X. Lu, “Molecular Behavior of Water on Titanium Dioxide Nanotubes: A Molecular Dynamics Simulation Study,” J. Chem. Eng. Data 2016, 61, 12, 4131-4138. (This reference should be given when the authors discuss the interaction of different models of water with TiO2 nanotubes);

b.      M. Matsui, M. Akaogi, “Molecular dynamics simulation of the structural and physical properties of the four polymorphs of TiO2,” Molecular Simulation, 1991, Vol. 6, pp. 239-244. (This reference should appear in the first place in line 102, where the authors address to the MA potential).

2.      It would be interesting to add some discussion about different empirical models describing the interactions in titanium dioxide. The Lennard-Jones reparameterization of the Buckingham potential could be used, but the authors abstained from it. Why? Is it so bad?

3.       L. 108, Table 1: not rij, but ρij. Moreover, the explanation of the Coulomb-Buckingham potential coefficients should be given in the text. What do they represent?

4.      L. 57: The authors state that “the computational time often scales as the cube of the number of electrons” and give the ref. [1] (Chen, X. & Mao, S.; Chemical Reviews. 107, 514, 2891-2959 (2007)). However, I’ve looked through the source under the ref. [1] and did not manage to find any statement concerning the computational costs. Maybe, I’ve missed something, but it is better if the authors check this reference once again and find this statement. If I really did miss this info, please, provide the page number of the statement about the computational costs of ref. [1] in the answer to me. If the authors have mistaken, they should provide the correct reference in the text.

5.      L82-83. In these lines the authors announce what they are going to consider in the current work: “In this study several aspects will be addressed, such as the density of vibrational states, the radial distribution function, the infrared spectrum and the effects of the interaction of different models of water with TiO2 nanotubes.” However, it is strange not see the discussion of the previous works devoted to these issues. Such works are plenty in the world literature. Obviously, the authors did not carry out the analysis of literature sources and stayed unaware of modern extensive computational studies of TiO2 nanotubes. I recommend to make the analysis of literature and write a comprehensive substantiation of their work. First of all, the authors should give the answers to the following questions: “Does their work address a specific gap in the field?”, “What does their work add to the subject area compared with other published material?”

6.      What is the Ewald summation method? The authors should give at least a couple of words to describe how this simulates the Coulomb interactions. The reference is mandatory.

7.      L. 200, 202, 230: Replace “,” with “.”

8.      L 206: Some strange misprints: “.to be aware of as well It is discovered …”

9.      L. 208: What does “f” mean in “(f 6 THz)”?

10.  L. 238: Replace “As far as …” with “as far as …”

11.  L. 271: What do the authors mean by “quantum observations”

12.  L. 281: Replace “We took into …” with “we took into …”

13.  L. 290: Replace “Hence„ these …” with “Hence, these …”

14.  L. 303-304: Some inconsistent phrase: “… the material of their study is irrelevant to our interest, but the effect of phonon softening is, While they have 5-layer graphene and consider it 2D in nature, their study shows that the top layer has more conformational …”

15.  I’m not going to criticize the essence, but the writing is awful! Can the authors be more eloquent in section 3.2?

16.  L. 308: Could the authors explain the connection? Why they suppose that having more conformational modes in the nanosheet structure as compared to the nanotube leads to the fact that surface phonons in the unwrapped nanosheets of anatase and rutile can be greater than those in the wrapped nanotube? It would be interesting for a common reader …

17.  L. 317: Replace “thatmeans” with “that means”. Check the whole phrase. It is not stylistically correct.

18.   L. 346: A misprint: “(see Fig. ??).”

19.  L. 351: “germanium motions are involved”? What do the authors write about? It is titanium nanotubes that are discussed. Please rewrite this paragraph, nothing is clear.  

20.  L. 443-450: Why “TiO” and not “TiO2”?

21.  L. 443-502: The conclusions are not consistent, do not clearly reflect the results obtained.

22.   L. 453-460 are superfluous, because these do not represent a conclusion but a description of the methodology used.

23.  L. 465: What a strange phrase “distinctiveness in distinguishing.” Rephrase the sentence.

24.  L. 473-475: What do the author mean by “molecular systems of TiO2 are stable for the Matsui and Akaogi (MA potential) for various Temperature and Molecular dynamics ensemble techniques.” Maybe, it better to write “Molecular dynamics simulations showed that molecular systems of TiO2 are stable?”

25.  L. 476, 489, 492, 493: “As far as I’m aware …”  Maybe, “As far as we are aware …”. “I found …”, “I have also observed …”, “I verified …” – the same.

Extensive revision of English is required.

Round 2

Reviewer 2 Report

The authors have followed all my recommendations. Now I recommend this manuscript for publication.